

# High metabolic load distance in professional soccer according to competitive level and playing positions

Tomás García-Calvo[1], José Carlos Ponce-Bordón[1], Eduard Pons[2], Roberto López del Campo[3], Ricardo Resta[3] and Javier Raya-González[4]

[1] Faculty of Sport Sciences, University of Extremadura, Cáceres, Spain
[2] Sports Performance Area, FC Barcelona, Barcelona, Spain
[3] Sport Research Section, LaLiga, Madrid, Spain
[4] Faculty of Health Sciences, Universidad Isabel I, Burgos, Spain

## ABSTRACT

**Background:** High metabolic load distance provides global information about the soccer players' total high-intensity activities. Thus, this study aimed to examine the Spanish professional soccer players' high metabolic load distance profile, comparing competitive level and playing positions.

**Methods:** A total of 18,131 individual match observations were collected from outfield players competing during the 2018/2019 and 2019/20 seasons in the First and Second Spanish Professional Soccer Leagues (LaLiga™). High Metabolic Load Distance (HMLD; distance covered with a power consumption above 25.5 W·kg$^{-1}$ and accelerations or decelerations (*e.g.*, accelerating from 2 to 4 m·s$^{-2}$ for 1 s) were included), and HMLD per minute (HMLD$_{min}$) were analyzed by the ChryonHego® video-tracking system. Players were classified according to their playing position as follows: Central Backs (CB), Full Backs (FB), Center Midfields (CM), Wide Midfields (WM), and Forwards (FW).

**Results:** No differences between competitive levels were found in any variable when all players were analyzed conjointly except for HMLD$_{min}$ overall and during the second half. However, when playing positions were considered, differences between competitive levels were observed in all positions, mainly in HMLD and HMLD during the first-half variables. In addition, several differences between playing positions were observed, with CB presenting the lowest values in all variables compared to their counterparts in both competitive levels, whereas CM in First Division and WM in Second Division showed the highest values in the HMLD variables.

**Discussion:** The findings are of interest to analyze the HMLD in professional soccer players, enabling the adaptation and individualization of training in this population according to the competitive level and specific playing position of each player.

Corresponding author
José Carlos Ponce-Bordón,
joponceb@unex.es

## INTRODUCTION

Match physical demands monitoring process aims to determine soccer players' demands during their match practice to replicate them during training sessions and, consequently, optimize their preparation (*Castillo et al., 2019*; *de Dios-Álvarez et al., 2021*; *Rojas-Valverde et al., 2019*). In this vein, the emergence of player tracking technologies has facilitated access to knowledge related to the external demands of professional soccer matches (*Cummins et al., 2013*). Specifically, the Global Position System (GPS) is considered a valid instrument to record external load demands in elite soccer during training and matches (*Mallo et al., 2015*). However, different professional leagues use different tracking systems, and not all of them have been validated against a gold-standard instrument (*Castellano, Alvarez-Pastor & Bradley, 2014*). In addition, these devices have presented problems related to the signal reception and can be uncomfortable for soccer players (*Hennessy & Jeffreys, 2018*). An interesting alternative to solve GPS-derived problems is the video camera tracking system, which has shown an almost perfect agreement with GPS (*Pons et al., 2019*). This system is composed of eight super cameras placed strategically throughout each stadium and allows calculating the positions for each player on several axes (*Pons et al., 2019*). Prior studies conducted in soccer have focused on reporting distances covered using fixed absolute velocity-based thresholds (*e.g.*, sprinting: distance covered above 24.0 km·h$^{-1}$; *Bush et al., 2015*; *Malone et al., 2015*). However, the intermittent nature of soccer match-play implies that analyses based only on velocity thresholds may be incomplete, and it is necessary to consider accelerations/decelerations and the energy cost of changing velocities (*Young et al., 2019*).

Metabolic power is a theoretical model to estimate the energy cost of acceleration and deceleration in team sports, including soccer (*Hoppe et al., 2017*; *Osgnach et al., 2010*). It proposes that accelerated running on level ground is energetically equivalent to running uphill at a constant speed (*de Hoyo et al., 2018*; *di Prampero et al., 2005*). In this regard, very high correlations between aerobic fitness variables and metabolic power estimations of high-power distance during professional matches have been found (*Manzi, Impellizzeri & Castagna, 2014*). However, metabolic power involves all actions, without discriminating them according to their intensity, although the relationship between high-intensity actions and soccer success has been proven (*Faude, Koch & Meyer, 2012*; *Yang et al., 2018*). In this context, a novel variable called high metabolic load distance (HMLD) is considered (*Tierney et al., 2016*). This variable refers to the distance covered with a power consumption above 25.5 W·kg$^{-1}$ (*Tierney et al., 2016*). Specifically, this value corresponds to running at a constant velocity of 5.5 m·s$^{-1}$ or 19.8 km·h$^{-1}$ on grass. Accelerations or decelerations (*e.g.*, accelerating from 2 to 4 m·s$^{-2}$ for 1 s) also are included in HMLD. This variable provides global information about the players' total high-intensity activities, as it not only includes high velocity running but also accelerations and decelerations, justifying its analysis in soccer (*Martín-García et al., 2018*).

Recent studies have been conducted focusing on external demands in professional soccer players including HMLD variables. Specifically, *Smpokos, Mourikis & Linardakis (2018)* analyzed the seasonal changes of the external demands in professional Greek soccer

players. These authors reported a maximum value of 1,880 m of HMLD in the most demanding phase of the season (January). On the other hand, *Tierney et al. (2016)* compared the external demands in professional soccer players according to their playing formation. These authors reported maximum HMLD values of 2,025.0 ± 304 m when 3-5-2 playing formation was used, presenting significantly lower values when other formations were selected (*e.g.*, 1,568.0 ± 257 m with 4-4-2 playing formation). In addition, these authors also reported that forwards achieved the highest HMLD values (2,476 ± 1,339 m) compared to their counterparts. *Martín-García et al. (2018)* analyzed the most demanding passages of play in soccer matches of professional players and observed values of MHLD per min of 26 ± 6.1 m in 10-min passages. Additionally, they found that wide midfields presented the greatest HMLD/min values. Despite these studies, specific knowledge about HMLD variables is scarce, as the aforementioned studies are relatively old, so the demands of HMLD could be different because the current demands, mainly related to high-intensity actions, have increased in recent years (*Pons et al., 2021*). Furthermore, there are no studies that combine the analysis of playing positions and levels.

To address the gap identified in the literature, we analyzed the external demands, in terms of HMLD, of Spanish male professional soccer players during two competitive seasons (from 2018/19 to 2019/20). Specifically, this study aimed to examine the Spanish professional soccer players' HMLD profile, comparing competitive level and playing positions. Based on previous studies (*Martín-García et al., 2018*; *Tierney et al., 2016*), we hypothesized that forwards and wide midfields would achieve the highest values of HMLD, whereas higher HMLD values would be observed during First Division matches.

## MATERIALS AND METHODS

### Participants

A total of 35,208 individual match observations of 1,666 professional soccer players were included in the study. Players belonged to all soccer teams that competed in the First and Second Division of the Spanish professional soccer league (LaLiga™) during the 2018/2019 and 2019/20 seasons. Goalkeepers and players who did not play the full game were excluded from the analysis. Thus, 17,077 recordings were excluded due to inclusion criteria, issues related to repeated signal loss by the system, or adverse weather conditions (*i.e.*, a match day too much rainy, windy, or snowy) during the match that hindered accurate data collection, so 18,131 match observations were analyzed (of 1,321 players). Players were classified according to their playing position as follows: Central Backs (CB), Full Backs (FB), Center Midfields (CM), Wide Midfields (WM), and Forwards (FW) (*Raya-González et al., 2020*). Data were provided to the authors by LaLiga™, which had informed all participants through its protocols. Written informed consent was obtained from all the participants and from a parent and/or legal guardian for subjects under 18. All data were anonymized according to the Declaration of Helsinki to ensure players' and teams' confidentiality, and the protocol was fully approved by the Ethics Committee of the University of Extremadura; Vice-Rectorate of Research, Transfer and Innovation—Delegation of the Bioethics and Biosafety Commission (Protocol number: 239/2019).

## Study design and procedures

According to previous studies (*Martín-García et al., 2018*), a retrospective, quasi-experimental longitudinal design was performed to examine the Spanish professional soccer players' HMLD profile, comparing competitive level and playing positions. Data were collected during 1,600 official matches across two full competition seasons (*i.e.*, 2018/2019 and 2019/2020), resulting in 18,131 match observations being collected. All the study variables were collected through the optical tracking system ChyronHego® (TRACAB, New York, NY, USA), which is composed of eight super 4K-High Dynamic Range cameras based on a positioning system (Tracab – ChyronHego VTS). This system records from several angles and analyzes X and Y positions for each player, providing real-time two-dimensional tracking (acquisition frequency = 25 Hz). The validity and reliability of the Tracab® video tracking system have been analyzed, reporting average measurement errors of 2% for the physical performance (*Linke, Link & Lames, 2018*). In addition, recent studies have tested the agreement between the Mediacoach® system and GPS devices (*Pons et al., 2021*), obtaining intraclass correlation coefficients (ICC) higher than 0.90.

## Variables

All the variables included were recorded by Mediacoach®, with the High Metabolic Load Distance (HMLD) as the main one. This variable refers to the distance covered with a power consumption above 25.5 W·kg$^{-1}$. This value corresponds to running at a constant velocity of 5.5 m·s$^{-1}$ or 19.8 km·h$^{-1}$ on grass. Accelerations or decelerations (*e.g.*, accelerating from 2 to 4 m·s$^{-2}$ for 1 s) also are included in HMLD. This variable provides global information on the players' total high-intensity activities, as it not only includes high velocity running, but also accelerations and decelerations. Specifically, relative values of HMLD were calculated: HMLD during the first half of the matches (HMLD1); HMLD during the second half of the matches (HMLD2); HMLD per minute (HMLD$_{min}$); HMLD during the first half of the matches per minute (HMLD1$_{min}$); and HMLD during the second half of the matches per minute (HMLD2$_{min}$).

## Statistical analysis

Data were analyzed using R-studio for Windows (*Rstudio, 2020*). A Linear Mixed Model (LMM) analysis was carried out for each of the eight models using the MIXED procedure. LMM allows analyzing data with a hierarchical structure in nested units and has demonstrated its ability to manage unbalanced and repeated-measures data (*Heck & Thomas, 2020*). For example, the distance covered in matches is nested for players across time (*i.e.*, each player has a record for any match played). To determine the adequacy of this statistical procedure, we first calculated the levels of within-person variance for each player by constructing unconditional null models. These unconditional models allowed us to calculate the Intraclass Correlation Coefficient (ICC), which presented values greater than 10%, indicating the existence of variability in the data and justifying this analysis approach (*Hox, Moerbeek & Van de Schoot, 2017*). Subsequently, some separate random intercept models were constructed for each outcome measure, with periods included as
**Table 1 Intercepts and standard error from high metabolic load distance profile attending to competitive level.**

| Variables | Total | First division | Second division | Comparison | $r^2$ |
|---|---|---|---|---|---|
| HMLD (m) | 2,347 (10.9) | 2,350 (12.7) | 2,344 (13.0) | 0.668 | 0.64 |
| HMLD1 (m) | 1,212 (5.8) | 1,218 (6.8) | 1,206 (7.2) | 0.104 | 0.62 |
| HMLD2 (m) | 1,134 (5.4) | 1,132 (6.5) | 1,134 (6.6) | 0.741 | 0.54 |
| $HMLD_{min}$ (m·min$^{-1}$) | 26.5 (0.1) | 26.7 (0.1) | 26.2 (0.1) | 0.018* | 0.61 |
| $HMLD1_{min}$ (m·min$^{-1}$) | 26.3 (0.1) | 26.4 (0.1) | 26.2 (0.1) | 0.221 | 0.63 |
| $HMLD2_{min}$ (m·min$^{-1}$) | 25.7 (0.1) | 25.9 (0.1) | 25.5 (0.2) | 0.043* | 0.55 |

Note:
HMLD, High Metabolic Load Distance; HMLD1, High Metabolic Load Distance during the first half of the matches; HMLD2, High Metabolic Load Distance during the second half of the matches; $HMLD_{min}$, High Metabolic Load Distance per minute; $HMLD1_{min}$, High Metabolic Load Distance during the first half of the matches per minute; $HMLD2_{min}$, High Metabolic Load Distance during the second half of the matches per minute.
* Differences between divisions (significant level was set at $p < .05$).

fixed effects. In this way, we compared the values of the variables obtained in First Division *vs* Second Division matches, or differences between playing positions. Finally, we provided the conditional $r^2$ metrics as a measure of effect size for each linear mixed model. Conditional $r^2$ concerned with variance explained by the whole model. The level of significance was set to 0.05.

# RESULTS

Table 1 presents the information about the HMLD variables and the comparison of competitive levels (*i.e.*, First division *vs* Second division). No significant differences between competitive levels were found for any variable except for $HMLD_{min}$ and $HMLD_{min2}$. Conditional $r^2$ values ranged from 0.54 to 0.64 in all models showing a moderate effect of competitive level on match physical demands.

Differences according to playing positions are shown in Tables 2, 3 and Fig. 1. CB presented the lowest values in all variables compared to their counterparts in both levels ($p < 0.001$). Significant differences were also observed when First Division CB were compared with Second Division CB in the HMLD, HMLD1, and $HMLD1_{min}$ variables ($p < 0.001$). Regarding FB, significant differences with CM were observed in all variables ($p < 0.05$) except for the HMLD2 in the Second Division. Differences between FB and WM were also observed in all variables ($p < 0.05$) except for HMLD and HMLD2 in the First Division. Differences between FB and FW ($p < 0.05$) were found in HMLD, HMLD1 (only in the First Division), HMLD2, $HMLD1_{min}$, and $HMLD2_{min}$ (only in the Second Division). Moreover, significant differences were observed when FW in the First Division were compared with the FW in the Second Division in HMLD and HMLD1 variables ($p < 0.05$). CM presented significant differences ($p < 0.05$) with WM in all variables except for $HMLD_{min}$ and $HMLD2_{min}$, whereas differences in $HMLD1_{min}$ were observed only in the First Division. Significant differences between CM and FW were observed in all variables and levels ($p < 0.05$). In all variables, CM in the First Division presented higher values compared to CM in the Second Division ($p < 0.001$). Attending to WM, significant differences with FW were found in all variables and levels ($p < 0.01$). Significant differences

**Table 2 Intercepts and standard error from high metabolic load distance profile attending to playing position.**

| Level | | First division | | | | | Second division | | | | | Comparisons | | | | | $r^2$ |
|---|---|---|---|---|---|---|---|---|---|---|---|---|---|---|---|---|---|
| Position | | CB | FB | CM | WM | FW | CB | FB | CM | WM | FW | CB | FB | CM | WM | FW | |
| HMLD (m) | | 1,964 (23.7)bcde | 2,434 (23.6)ace | 2,586 (22.3)abde | 2,464 (27.3)ace | 2,290 (31.5)abcd | 2,050 (23.8)bcde | 2,382 (24.5)acde | 2,473 (20.3)abde | 2,542 (29.8)abce | 2,296 (32.4)abcd | 0.00*** | 0.05* | 0.00*** | 0.01* | 0.88 | 0.64 |
| HMLD1 (m) | | 1,013 (12.7)bcde | 1,246 (12.7)acde | 1,351 (12.0)abde | 1,287 (14.8)abce | 1,196 (16.9)abcd | 1,065 (12.8)bcde | 1,211 (13.1)acd | 1,282 (10.9)abe | 1,293 (16.1)abe | 1,185 (17.5)acd | 0.00*** | 0.02* | 0.00*** | 0.74 | 0.60 | 0.62 |
| HMLD2 (m) | | 960 (12.0)bcde | 1,189 (12.0)ace | 1,234 (11.4)abde | 1,183 (14.2)ace | 1,094 (15.9)abcd | 979 (12.1)bcde | 1,168 (12.4)ade | 1,196 (10.3)ade | 1,240 (15.4)abce | 1,108 (16.5)abcd | 0.18 | 0.13 | 0.00*** | 0.00*** | 0.51 | 0.54 |

**Notes:**

HMLD, High Metabolic Load Distance; HMLD1, High Metabolic Load Distance during the first half of the matches; HMLD2, High Metabolic Load Distance during the second half of the matches; HMLD$_{min}$, High Metabolic Load Distance per minute; HMLD1$_{min}$, High Metabolic Load Distance during the first half of the matches per minute; HMLD2$_{min}$, High Metabolic Load Distance during the second half of the matches per minute; CB, Central Backs; FB, Full Backs; CM, Center Midfields; WM, Wide Midfields; FW, Forwards.

a significative differences compared to Central Backs.
b significative differences compared to Full Backs.
c significative differences compared to Center Midfields.
d significative differences compared to Wide Midfields.
e significative differences compared to Forwards.
* Differences between divisions (significant level was set at $p < .05$);
*** Differences between divisions (significant level was set at $p < .001$).

**Table 3 Intercepts and standard error from relative high metabolic load distance profile attending to playing position.**

| Level | First division | | | | | Second division | | | | | Comparisons | | | | | $r^2$ |
|---|---|---|---|---|---|---|---|---|---|---|---|---|---|---|---|---|
| Variables | CB | FB | CM | WM | FW | CB | FB | CM | WM | FW | CB | FB | CM | WM | FW | |
| HMLD$_{min}$ (m·min$^{-1}$) | 21.3 (0.3)[bcde] | 26.2 (0.3)[acd] | 29.3 (0.2)[abde] | 28.4 (0.3)[abce] | 26.9 (0.3)[acd] | 21.8 (0.3)[bcde] | 26.0 (0.3)[acd] | 28.0 (0.2)[abe] | 28.1 (0.3)[abe] | 26.1 (0.3)[acd] | 0.07 | 0.48 | 0.00 *** | 0.26 | 0.05* | 0.61 |
| HMLD1$_{min}$ (m·min$^{-1}$) | 21.7 (0.3)[bcde] | 26.7 (0.3)[acd] | 29.0 (0.2)[abde] | 27.8 (0.3)[abce] | 26.3 (0.3)[acd] | 22.9 (0.3)[bcde] | 26.3 (0.3)[acd] | 27.9 (0.2)[abe] | 27.9 (0.3)[abe] | 25.4 (0.3)[abcd] | 0.00*** | 0.22 | 0.00 *** | 0.88 | 0.02* | 0.63 |
| HMLD2$_{min}$ (m·min$^{-1}$) | 20.4 (0.3)[bcde] | 25.2 (0.3)[acde] | 28.3 (0.2)[abe] | 28.0 (0.3)[abe] | 26.2 (0.3)[abcd] | 20.6 (0.3)[bcde] | 25.1 (0.3)[acd] | 27.1 (0.2)[abe] | 27.7 (0.3)[abe] | 25.8 (0.3)[acd] | 0.66 | 0.80 | 0.00 *** | 0.36 | 0.32 | 0.55 |

**Notes:**

HMLD, High Metabolic Load Distance; HMLD1, High Metabolic Load Distance during the first half of the matches; HMLD2, High Metabolic Load Distance during the second half of the matches; HMLD$_{min}$, High Metabolic Load Distance per minute; HMLD1$_{min}$, High Metabolic Load Distance during the first half of the matches per minute; HMLD2$_{min}$, High Metabolic Load Distance during the second half of the matches per minute; CB, Central Backs; FB, Full Backs; CM, Center Midfields; WM, Wide Midfields; FW, Forwards.

[a] significative differences compared to Central Backs.
[b] significative differences compared to Full Backs.
[c] significative differences compared to Center Midfields.
[d] significative differences compared to Wide Midfields.
[e] significative differences compared to Forwards.
* Differences between divisions (significant level was set at $p < .05$);
*** Differences between divisions (significant level was set at $p < .001$).

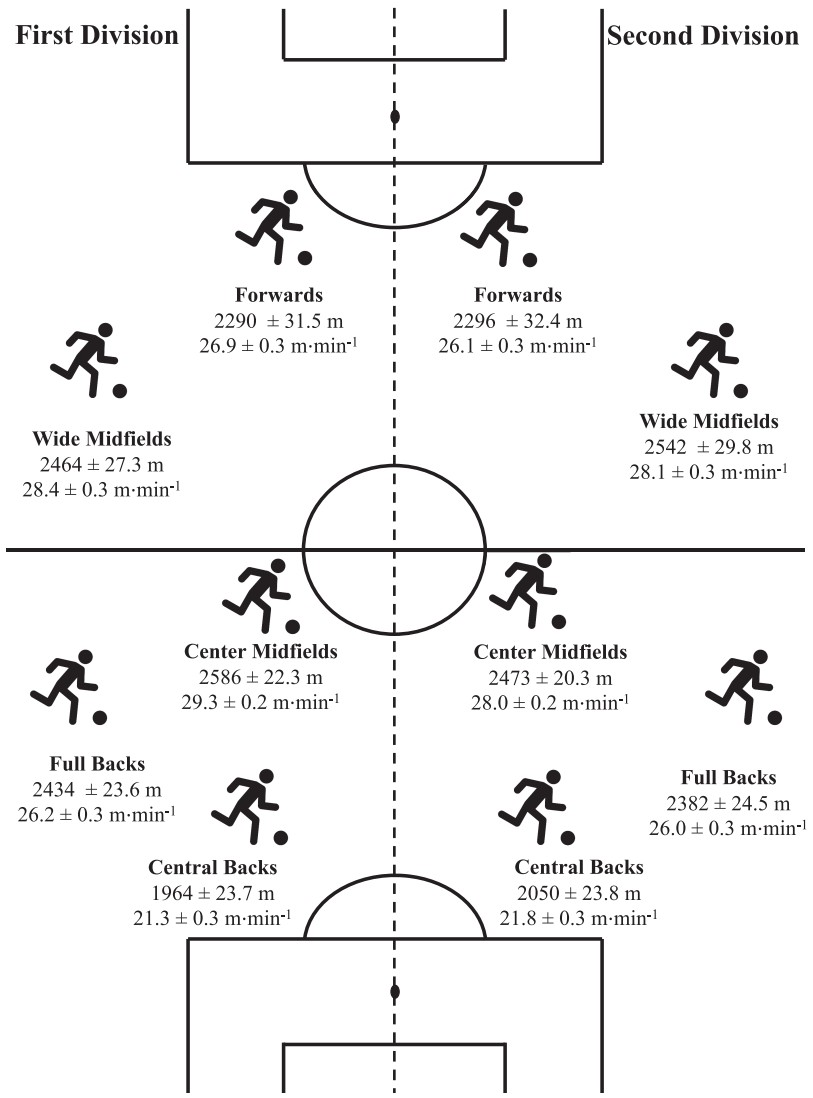

**First Division**

**Second Division**

Forwards
2290 ± 31.5 m
26.9 ± 0.3 m·min⁻¹

Forwards
2296 ± 32.4 m
26.1 ± 0.3 m·min⁻¹

Wide Midfields
2464 ± 27.3 m
28.4 ± 0.3 m·min⁻¹

Wide Midfields
2542 ± 29.8 m
28.1 ± 0.3 m·min⁻¹

Center Midfields
2586 ± 22.3 m
29.3 ± 0.2 m·min⁻¹

Center Midfields
2473 ± 20.3 m
28.0 ± 0.2 m·min⁻¹

Full Backs
2434 ± 23.6 m
26.2 ± 0.3 m·min⁻¹

Full Backs
2382 ± 24.5 m
26.0 ± 0.3 m·min⁻¹

Central Backs
1964 ± 23.7 m
21.3 ± 0.3 m·min⁻¹

Central Backs
2050 ± 23.8 m
21.8 ± 0.3 m·min⁻¹

**Figure 1 Position-specific high metabolic load distance profiles.**

between WM in both levels were also found in HMLD and HMLD2 ($p < 0.05$). Significant differences between FW in both levels were only observed in $HMLD1_{min}$ ($p < 0.05$). Finally, conditional $r^2$ values ranged from 0.54 to 0.64 in all models showing a moderate effect of playing position on match physical demands.

## DISCUSSION

The main aim of this study was to examine Spanish professional soccer players' HMLD profile, comparing competitive level and playing positions. This is the first research that compares the match physical demands of competitive level and playing positions attending to this variable, which reflects the high-intensity profile of soccer players. No differences between competitive levels were found in any variable when all players were analyzed

conjointly except for $HMLD_{min}$ and $HMLD_{min2}$. However, when playing positions were considered, differences between competitive levels were observed in all positions, mainly in the HMLD and HMLD1 variables. Several differences between playing positions were also observed, with CB presenting the lowest values in all variables compared to their counterparts in both competitive levels, whereas CM in the First Division and WM in the Second Division showed the highest values in the HMLD variables.

This novel variable has gained relevance in recent years but the research literature has revealed few studies with professional soccer players. In our study, mean values of 2,347 ± 10.9 m were found when both competitive levels were considered conjointly. These values are higher than those reported by *Tierney et al. (2016)* and *Smpokos, Mourikis & Linardakis (2018)*, who analyzed England sub-23 league and professional Greek soccer players in official matches and presented HMLD values of 2,025.0 ± 304 m and 1,880 m, respectively. These differences could be because the aforementioned studies are older than ours. Also, several authors have recently concluded that the distance covered at high intensity is increasing in elite soccer (*Pons et al., 2021*). Additionally, non-significant differences were observed when the first half was compared to the second half, similar to the report of *Casamichana et al. (2019)*, which is justified by the fact that lower distances at high-intensity are covered by soccer players during second halves in official matches (*Pons et al., 2021*; *Rivilla-García et al., 2019*). Finally, regarding the comparison of match physical demands between professional soccer leagues, no significant differences were found when the First and Second Division were compared attending to HMLD variables, except for $HMLD_{min}$ and $HMLD_{min2}$. On the contrary, previous research have showed differences in match physical demands between soccer leagues. For example, the Premier League soccer players covered less total distance and high intensity running distances than the lower leagues players (*Bradley et al., 2013*; *Di Salvo et al., 2013*). In addition, recent studies have analyzed the Spanish professional soccer leagues with the same aim, and they concluded that there are significant differences, finding that the top-tiered leagues were more physically demanding than the lower (*Castellano et al., 2015*; *Gomez-Piqueras et al., 2019*; *Pons et al., 2021*). However, our results could be influenced by considering all the players conjointly, without differentiating playing positions.

Previous studies have determined that professional soccer players' external demands during match-play vary according to their specific playing position (*Martín-García et al., 2018*). Therefore, it is necessary to analyze HMLD considering this differentiation for a deeper knowledge of the match physical demands of the competition. This would allow optimizing the decision-making process for the prescription and manipulation of the training load of professional soccer players. In this study, differences in HMLD were observed according to playing positions, with the CB players presenting the lowest HMLD values in all the variables, in both the First and Second Division. These results support those found by *Martín-García et al. (2018)*, who observed that CB players had the lowest HMLD values (*i.e.*, m·m$^{-1}$) in all the passages analyzed (*i.e.*, 1, 3, 5 and 10 min) compared to their counterparts. *Tierney et al. (2016)* also obtained similar results in CB players. This

consensus seems to be due to these players' predominantly defensive roles on the field (*e.g.*, situations near to the goal without longer distances covered at high intensity), which favors their HMLD values being significantly lower than those of the rest of the playing positions. On the other hand, the highest HMLD distances are covered by different playing positions as a function of competitive level, with CM players obtaining the highest values in the First Division, whereas the highest values in the Second Division were observed in WM players. These differences between competitive levels could be related to the differences in the playing styles followed in each division (*Pons et al., 2021*). However, our results disagree with those obtained in the study conducted by *Tierney et al. (2016)*, in which FW presented the highest HMLD values. When the most demanding passages of match-play are analyzed (*Martín-García et al., 2018*), WM presented the highest values. These players could be involved in the most determinant actions in soccer, such as goals, which are characterized by the presence of sprints (*Faude, Koch & Meyer, 2012*; *Yang et al., 2018*). Finally, when playing positions and competitive levels interacted, differences between competitive levels were observed in all positions, mainly in HMLD and HMLD1 variables. These findings suggest that it is necessary to develop an individualized profile attending to the interaction of positions and competitive levels and thus achieve individualized training tasks to meet the demands of each player.

This study is not without limitations. Firstly, this investigation was conducted only in the top two Spanish professional soccer leagues, so the findings obtained cannot be extrapolated to different leagues, as external demands vary according to the country. Secondly, the teams were not differentiated according to different contextual variables, such as the match location, the quality of the opponent, or the playing style. In addition, would be interesting considered the match status (*Andrzejewski et al., 2016*) and the evolution of score-line (*Redwood-Brown et al., 2012*), since research have showed their influence on position-specific match physical demands, showing that central defenders and fullbacks covered shorter distances at high intensity in won matches than in lost matches. Finally, it would also be appropriate to differentiate between teams of high and low competitive density because considering them conjointly could influence the results.

## CONCLUSIONS

In conclusion, this study has confirmed the lack of HMLD differences between competitive levels when all the players were considered conjointly, except for $HMLD_{min}$ and $HMLD_{min2}$. Differences between competitive levels were obtained in all positions when analyzed isolatedly, mainly in the HMLD and HMLD1 variables. Several differences between playing positions were also observed, with CB presenting the lowest values in all variables compared to their counterparts in both levels, whereas CM in the First Division and WM in the Second Division obtained the highest values in the HMLD variables. In practical terms, this study could serve as a reference for the analysis of HMLD in professional soccer players, improving the adaptation and individualization of the training process in this profile of team sport athletes according to the competitive level and specific playing position of each player.

### Funding

This work was supported by the European Regional Development Fund (ERDF), the Government of Extremadura (Department of Economy and Infrastructure) and LaLiga Research and Analysis Sections. The funders had no role in study design, data collection and analysis, decision to publish, or preparation of the manuscript.

### Grant Disclosures

The following grant information was disclosed by the authors:
European Regional Development Fund.
Government of Extremadura (Department of Economy and Infrastructure).
LaLiga Research and Analysis Sections.

### Competing Interests

The authors declare that they have no competing interests. Eduard Pons is employed by FC Barcelona. Roberto Lopez del Campo & Ricardo Resta are employed by LaLiga.

### Author Contributions

- Tomás García-Calvo conceived and designed the experiments, performed the experiments, analyzed the data, authored or reviewed drafts of the article, and approved the final draft.
- José Carlos Ponce-Bordón conceived and designed the experiments, prepared figures and/or tables, authored or reviewed drafts of the article, and approved the final draft.
- Eduard Pons conceived and designed the experiments, authored or reviewed drafts of the article, and approved the final draft.
- Roberto López del Campo conceived and designed the experiments, authored or reviewed drafts of the article, funding and data providing, and approved the final draft.
- Ricardo Resta conceived and designed the experiments, authored or reviewed drafts of the article, funding and data providing, and approved the final draft.
- Javier Raya-González conceived and designed the experiments, performed the experiments, analyzed the data, authored or reviewed drafts of the article, and approved the final draft.

### Ethics

The following information was supplied relating to ethical approvals (*i.e.*, approving body and any reference numbers):

All data were anonymized according to the Declaration of Helsinki to ensure players' and teams' confidentiality, and the protocol was fully approved by the Ethics Committee of the University of Extremadura; Vice-Rectorate of Research, Transfer and Innovation – Delegation of the Bioethics and Biosafety Commission (Protocol number: 239/2019).

## Data Availability

The raw data are available in the Supplemental File.

## Supplemental Information

Supplemental information for this article can be found online at http://dx.doi.org/10.7717/peerj.13318#supplemental-information.

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
