# Peer review of "High metabolic load distance in professional soccer according to competitive level and playing positions"

_PeerJ, doi:10.7717/peerj.13318_

## Round 0.1 · original submission · Minor Revisions

The reviewers agreed that the article has potential, but some changes must be conducted before acceptance.

·

Basic reporting

General Comments

I congratulate the authors for their efforts. This research has a really nice design and a large dataset. However, I have some doubts. I have stated my doubts and recommendations.

Experimental design

It is very nicely designed.

Validity of the findings

The findings are pretty good. Graphics just need to be added to better convey the findings to the readers.

·

Basic reporting

Dear Authors
-Thank you for your interesting research, generally, your manuscript is valuable for sports sciences and soccer. I think your paper needs some minor grammatical corrections. In addition, I will write some minor revisions below.

Abstract
-If it is possible please revise the background section. I think you can write more specific information about HMLD for soccer.
Introduction
-Line 77-81 It would be more appropriate if the two sentences in these lines were combined and written.

-Please add your main hypothesis at the end of the introduction

Materials & Method, Results and discussion no need any revisions.

-Please edit in English to make your article more readable.

Experimental design

Your paper experimental design is sufficient

Validity of the findings

Your paper's findings are so valuable

Additional comments

I have no additional comments

·

Basic reporting

Many thanks to the authors for this beautiful work.
-In the introduction, why only the HMLD profile was chosen, why the HMDL assessment is important and why it is important should be explained in the context of player position and league level. In other words, the importance of the study in terms of literature should be mentioned.

Experimental design

In the method section, the research design should be based on scientific sources.
Are the matches of all teams in the 1st and 2nd leagues analyzed in Laliga? Information must be given.
In the method section, adverse weather conditions? It should be explained.
How long the matches specified in the method section were analyzed and by whom. It should be explained.

Validity of the findings

The discussion section is very short. Not enough current resources have been used. This section should be written in detail for those who increase the number of references. In addition, the findings in the discussion section could not be concluded sufficiently. The reasons for the findings should be explained in detail. (For example, there is no difference in HMDL in the first and second leagues)

Additional comments

After corrections, the article can be published.

---

## Round 0.2 · accepted · Accept

Can be accepted in its current form.

·

Basic reporting

Congratulations to the authors for their efforts. If the article is suitable for reviewers and editors, it can be published.

Experimental design

Everything is fine.

Validity of the findings

Everything is fine.

Additional comments

Everything is fine.

·

Basic reporting

Dear Author,

Thank you for your effort, I think your manuscript is ready to publish.

Experimental design

Dear Author,

Thank you for your effort, I think your manuscript is ready to publish.

Validity of the findings

Dear Author,

Thank you for your effort, I think your manuscript is ready to publish.

Additional comments

Dear Author,

Thank you for your effort, I think your manuscript is ready to publish.

·

Basic reporting

I examined the rebuttal letter and also text carefully.. Authors improved the article.. In my opinion, it is ready for publication..

Experimental design

I examined the rebuttal letter and also text carefully.. Authors improved the article.. In my opinion, it is ready for publication..

Validity of the findings

I examined the rebuttal letter and also text carefully.. Authors improved the article.. In my opinion, it is ready for publication..

Additional comments

I examined the rebuttal letter and also text carefully.. Authors improved the article.. In my opinion, it is ready for publication..